# Combination Therapy of Cuban Policosanol (Raydel^®^, 20 mg) and Intensive Exercise for 12 Weeks Resulted in Improvements in Obesity, Hypertension, and Dyslipidemia without a Decrease in Serum Coenzyme Q_10_: Enhancement of Lipoproteins Quality and Antioxidant Functionality in Obese Participants

**DOI:** 10.3390/ph17010132

**Published:** 2024-01-19

**Authors:** Kyung-Hyun Cho, Hyo-Seon Nam, Na-Young Kim, Myeong-Sung Lee, Dae-Jin Kang

**Affiliations:** Raydel Research Institute, Medical Innovation Complex, Daegu 41061, Republic of Korea; sun91120@raydel.co.kr (H.-S.N.); christy3206@raydel.co.kr (N.-Y.K.); hanpower24@raydel.co.kr (M.-S.L.); daejin@raydel.co.kr (D.-J.K.)

**Keywords:** high-density lipoproteins, apolipoprotein A-I, policosanol, exercise, paraoxonase, low-density lipoproteins, coenzyme Q_10_

## Abstract

Obesity and overweight, frequently caused by a lack of exercise, are associated with many metabolic diseases, such as hypertension, diabetes, and dyslipidemia. Aerobic exercise effectively increases the high-density lipoproteins-cholesterol (HDL-C) levels and alleviates the triglyceride (TG) levels. The consumption of Cuban policosanol (Raydel^®^) is also effective in enhancing the HDL-C quantity and HDL functionality to treat dyslipidemia and hypertension. On the other hand, no study has examined the effects of a combination of high-intensity exercise and policosanol consumption in obese subjects to improve metabolic disorders. In the current study, 17 obese subjects (average BMI 30.1 ± 1.1 kg/m^2^, eight male and nine female) were recruited to participate in a program combining exercise and policosanol (20 mg) consumption for 12 weeks. After completion, their BMI, waist circumference, total fat mass, systolic blood pressure (SBP), and diastolic blood pressure (DBP) reduced significantly up to around −15%, −13%, −33%, −11%, and −13%, respectively. In the serum lipid profile, at Week 12, a significant reduction was observed in the total cholesterol (TC) and triglyceride (TG) levels, up to −17% and −54% from the baseline, respectively. The serum HDL-C was elevated by approximately +12% from the baseline, as well as the percentage of HDL-C in TC, and HDL-C/TC (%), was enhanced by up to +32% at Week 12. The serum coenzyme Q_10_ (CoQ_10_) level was increased 1.2-fold from the baseline in all participants at Week 12. In particular, the male participants exhibited a 1.4-fold increase from the baseline. The larger rise in serum CoQ_10_ was correlated with the larger increase in the serum HDL-C (*r* = 0.621, *p* = 0.018). The hepatic function parameters were improved; the serum γ-glutamyl transferase decreased at Week 12 by up to −55% (*p* < 0.007), while the aspartate aminotransferase and alanine transaminase levels diminished within the normal range. In the lipoprotein level, the extent of oxidation and glycation were reduced significantly with the reduction in TG content. The antioxidant abilities of HDL, such as paraoxonase (PON) and ferric ion reduction ability (FRA), were enhanced significantly by up to 1.8-fold and 1.6-fold at Week 12. The particle size and number of HDL were elevated up to +10% during the 12 weeks, with a remarkable decline in the TG content, glycation extent, and oxidation. The improvements in HDL quality and functionality were linked to the higher survivability of adult zebrafish and their embryos, under the co-presence of carboxymethyllysine (CML), a pro-inflammatory molecule known to cause acute death. In conclusion, 12 weeks of Cuban policosanol (Raydel^®^, 20 mg) consumption with high-intensity exercise displayed a significant improvement in blood pressure, body fat mass, blood lipid profile without liver damage, CoQ_10_ metabolism, and renal impairment.

## 1. Introduction

Obesity and overweight are global health issues with the exacerbation of cardiovascular risk and metabolic syndrome, such as hypertension and dyslipidemia with low high-density lipoproteins (HDL)-cholesterol and high triglyceride (TG) [1,2]. Sedentary behaviors pose a considerable threat to the onset of metabolic syndrome and cardiovascular disease (CVD) [3,4]. These conditions often correlate with low HDL-C, elevated TG, insulin resistance, and obesity [5,6]. Physical activity serves as a therapeutic measure to decrease the possibility of CVD and overall mortality in a way affected by the intensity and duration of the exercise [7,8]. On the other hand, the optimal length of the treatment period and exercise intensity to achieve a desirable weight loss, normal blood pressure range, and improvement of serum lipid profile are still unclear, depending on age and gender.

The impact of exercise workouts on increasing HDL-C levels was disappointing: a 1.1 and 1.4 mg/dL elevation of HDL-C for men and women, respectively, from five months of aerobic exercise [9]. Although the correlation between the enhancement in HDL-C and the exercise period and exercise intensity is unclear, the increase in HDL-C levels was unsatisfactory, ranging from 0.27 to 5.41 mg/dL from 23 meta-analyses [10]. In addition, the blood pressure (BP) lowering effects of exercise were also unsatisfactory, varying between systolic BP (SBP) and diastolic BP (DBP). Aerobic and static exercise can reduce the SBP significantly, but not the DBP [11,12]. These findings indicate that exercise alone may not be adequate for achieving sufficient treatment efficiencies in lowering the BP and raising the HDL-C levels, even though it effectively reduced the body weight, body fat, and serum TG levels.

Although a combination of statin, either rosuvastatin or atorvastatin, and exercise therapy for 20 weeks raised the HDL-C significantly (from 46 ± 14 mg/dL to 54 ± 14 mg/dL), the combination therapy did not affect BP [13]. The use of statin has been closely associated with undesirable musculoskeletal damage, such as muscular pain, cramps, and stiffness [14]. A combination of statin and high-intensity exercise (eccentric and strenuous exercises) caused muscle injury, whereas a combination with moderate exercise did not cause myopathy [15]. On the other hand, many studies strongly suggest that statin-associated myopathy and myalgia were more exacerbated with an increase in exercise [16], which have been the most common statin side effects [17]. Myopathy and myalgia were more associated with the decrease in CoQ_10_ by statin consumption and regular exercise [18]. Therefore, augmenting or preserving the CoQ_10_ levels could be an indispensable strategy to treat dyslipidemia and atherosclerosis without adverse effects [19]. 

On the other hand, prehypertensive subjects who underwent Cuban policosanol supplementation for 8 weeks, 12 weeks, or 24 weeks experienced a substantial reduction in both SBP and DBP, along with an elevation in HDL-C levels. This effect was observed even without exercise training [20,21,22]. On the other hand, no study has compared the efficacy of treating hypertension and dyslipidemia by a combination of exercise and policosanol intake in obese subjects. The correlation between the change in BP and HDL-C levels after combination therapy of intense exercise and policosanol intake in obese individuals needs to be determined.

A nationally representative study with the Korea National Health and Nutrition Examination Survey indicated that patients with metabolic syndrome had significantly higher levels of liver enzymes, aspartate aminotransferase (AST), alanine transaminase (ALT), γ-glutamyl transferase (GTP), and alkaline phosphatase (ALP), than the control subjects [23]. In addition, liver function enzymes, AST, ALT, and γ-GTP, were usually elevated by high-intensity exercise, particularly in weight training, as an intense exercise-induced liver injury [24]. Therefore, it is necessary to compare the change in the serum enzymes for liver damage and CoQ_10_ level by a combination therapy of Cuban policosanol (20 mg/day) and high-intensity exercise in subjects with metabolic syndrome. 

The study aimed to assess the difference in blood pressure, lipid profile parameters, hepatic function parameters, and serum CoQ_10_ levels among obese participants after 12 weeks of consuming Cuban policosanol and following high-intensity exercise compared to their baseline values. The changes in the qualities of lipoproteins, very-low-density lipoproteins (VLDL), LDL, HDL_2_, and HDL_3_, were compared using composition analysis, the extent of oxidation and glycation, and transmission electron microscopy (TEM). As an HDL functionality test, the antioxidant and anti-inflammatory activities were compared using zebrafish embryo survivability under the presence of carboxymethyllysine (CML). Zebrafish embryos exhibit sophisticated innate and acquired immune systems like mammals [25]. One notable benefit of utilizing zebrafish embryos is their external development and optical transparency throughout the developmental stages. These unique features make zebrafish and their embryos highly valuable animal models for diverse research, such as investigating antioxidants and anti-inflammatory agents [26].

## 2. Results

### 2.1. Changes in Anthropometric Profiles 

The SBP and DBP reduced significantly by −7.7% (*p* = 0.046) and −12.8% (*p* = 0.007), respectively, at Week 12 of consumption of policosanol (PCO) plus exercise compared to the baseline (Week 0), suggesting that policosanol intake and exercise improved prehypertension (Table 1). In particular, the male group, which had more hypertension at Week 0, displayed a more remarkable decline in the SBP and DBP by up to −11% and −19% than those of the female group. All participants demonstrated a remarkable reduction in body weight, waist circumference, total fat mass, and body fat percentage during the 12 weeks of consumption, especially in male participants. On the other hand, there was no change in muscle mass and body water content during 12 weeks of consumption compared to the baseline value. 

### 2.2. Change in Serum Coenzyme Q_10_ and Lipid Profiles

After 12 weeks of policosanol intake and exercise, all participants displayed a remarkable decrease in the total cholesterol (TC), triglyceride (TG), and remnant cholesterol (RC) levels by up to −17%, −53%, and −49%, respectively. The HDL-C and HDL-C/TC (%) levels were also elevated by up to 12% and 31%, respectively (Table 2). Specifically, male participants exhibited a more substantial reduction in TC and TG, approximately −21% and −56%, respectively. Additionally, the male participants experienced a greater rise in HDL-C, about 24%, compared to their female counterparts. All participants exhibited a remarkable reduction in RC, TG/HDL-C, LDL-C, and HDL-C:LDL-C (L:H) ratio after 12 weeks, as listed in Table 2. 

Interestingly, the serum coenzyme Q_10_ (CoQ_10_) level of all participants was elevated 1.2-fold at Week 12 (492 ± 25 ng/mL) from the baseline (408 ± 30 ng/mL), as depicted in Figure 1A. In addition, the male participants displayed a significant elevation in the serum CoQ_10_, 1.4-fold (*p* = 0.027) higher at Week 12 (471 ± 27 ng/mL) than Week 0 (341 ± 28 ng/mL), while the female participants showed a similar level between Weeks 12 (513 ± 43 ng/mL) and 0 (476 ± 48 ng/mL). The findings indicate that the combination of policosanol intake and exercise had no impact on CoQ_10_ metabolism in either males or females. However, within the male group, there was a notable rise in serum CoQ_10_ levels after 12 weeks of consumption. Spearman correlation analysis revealed a significant positive correlation (*r* = 0.621, *p* = 0.018) between the net enhancement in CoQ_10_ levels and the net increase in HDL-C between Weeks 0 and 12, as illustrated in Figure 1B. No correlation was observed between the net change in TC, TG, and LDL-C and the net rise in CoQ_10_ levels.

The serum CoQ_10_/TC ratio was elevated by 45% (*p* = 0.034) at Week 12 from the baseline in the total participants (Table 2). In particular, the male group displayed a 73% rise (*p* = 0.007) in the CoQ_10_/TC ratio at Week 12 from the baseline. Moreover, the CoQ_10_/LDL-C ratio at Week 12 was approximately 51% (*p* = 0.026) higher than the baseline in the total participants; the male group showed an 82% enhancement (*p* = 0.007) in the CoQ_10_/LDL-C ratio. On the other hand, although the female group revealed a 28% and 34% boost in the CoQ_10_/TC ratio and CoQ_10_/LDL-C ratio, respectively, at Week 12 from the baseline, no notable significance was detected between Weeks 0 and 12. Interestingly, the CoQ_10_/HDL-C ratio was not changed in the male and female groups between Weeks 0 and 12, 370–415 ng/mL. These results suggest that the serum CoQ_10_ level was not depreciated by exercise and policosanol intake even though the TC, TG, and LDL-C decreased remarkably. At Week 12, the rise in the serum CoQ_10_ level correlated with the increase in HDL-C. 

### 2.3. Change in the Serum Protein Parameters 

All participants exhibited reduced serum aspartate transferase (AST), alanine aminotransferase (ALT), creatinine, and estimated glomerular filtration (e-GRF) levels, which were in the normal range during the 12 weeks of policosanol intake and exercise, suggesting no notable changes in hepatic damage and kidney impairment (Table 3). On the other hand, gamma-glutamyl transferase (γ-GTP) was reduced remarkably (by −55%; *p* = 0.007) in all participants at Week 12 from the baseline. In particular, the male participants demonstrated a larger reduction (up to −59%; *p* = 0.046) from the baseline. The serum glucose, apoA-I, apo-B level, apoA-I/apo-B ratio, and hsCRP levels did not change significantly between Weeks 0 and 12, which fell within the normal range (Table 3). These results indicate that policosanol intake and exercise induced significant improvement in the hepatic parameters without impairing the renal functions, apolipoproteins metabolism, and systemic inflammation. 

### 2.4. Change in VLDL and LDL Properties and Compositions

In very-low-density lipoprotein (VLDL), the extent of glycation and oxidation lowered remarkably (−42 to −43%) in the male and female groups, representing an approximately −26% and −44% decrease in the TC and TG content, respectively, during 12 weeks of consumption (Table 4). Interestingly, at Week 0, the female participants exhibited 30%, 12%, and 9% higher glycation extent, TC content, and TG content, respectively, than those of the male group in VLDL, while the oxidation extent was 34% lower than the male participants. Conversely, the female group experienced a more substantial reduction in glycation, oxidation, TC content, and TG content than the male group, resulting in an impressive intensification of the VLDL properties. These results suggest that policosanol intake and exercise improved the VLDL properties remarkably to be less atherogenic. 

By Week 12, there was a notable improvement in the properties of low-density lipoprotein (LDL) particles, marked by a substantial reduction in glycation and oxidation (up to −14 and −22%, respectively), as depicted in Table 4. In addition, the TC and TG content in LDL were alleviated by −32% and −43%, respectively, at Week 12 compared to the baseline. Despite the decrease in TC and TG contents, the particle diameter of LDL (25–27 nm) did not reduce at Week 12, suggesting that the LDL particle had changed to more healthy qualities with less oxidation and glycation. In the male participants, the LDL particle size was enhanced by 9% at Week 12 compared to the baseline, while no change was observed with the female participants, suggesting that the improvement was more distinct in the male groups. 

### 2.5. Electromobility of LDL and Extent of Oxidation

The intact charge and three-dimensional structure of LDL influence the electrophoretic mobility. Oxidized LDL with greater apo-B fragmentation exhibited faster migration toward the gel bottom, which is attributed to an increased negative charge. Enhanced LDL oxidation resulted in swifter electromobility in agarose gel because of the smaller particle size and apo-B fragmentation with a larger TG content in LDL particles. Therefore, the electrophoretic mobility pattern and the intensity of the separated bands offer valuable insights about LDL oxidation. As illustrated in Figure 2A, by the 12th week, all participants exhibited a declaration in electromobility, accompanied by an approximately 13% rise in band intensity (BI) compared to the baseline at Week 0. At Week 12, the BI of LDL in all participants was more distinct than a Week 0. In contrast, oxidized LDL showed an almost absent and smeared band intensity with the aggregation of LDL in the loading position, as indicated by the arrow. Quantitative analysis of oxidized species in all participants revealed a roughly −22% reduction in malondialdehyde (MDA) within LDL at Week 12 compared to the baseline (Figure 2B). Specifically, male participants exhibited a decrease of up to −32% in MDA, indicating a more significant improvement in LDL among the male group.

### 2.6. Electron Microscopic Observation of Lipoprotein Image

At Week 0, all lipoproteins demonstrated irregular size distributions and ambiguous particle morphologies with aggregation, as indicated by the red arrowhead in VLDL, LDL, HDL_2_, and HDL_3_ (Figure 3). On the other hand, at Week 12, all lipoproteins had a more regular size distribution and distinct particle morphology without aggregation. At Week 12, the particle numbers of VLDL and LDL depreciated, while the particle numbers of HDL_2_ and HDL_3_ increased, as depicted in Figure 3. 

During 12 weeks, although the VLDL and LDL particle sizes were relatively unchanged (1195–1364 nm^2^ (size) and 528–576 nm^2^ (size), respectively), the HDL_2_ particle size enhanced significantly up to 10% (*p* = 0.032) from the baseline (127 ± 5 nm^2^ and 139 ± 3 nm^2^ for Weeks 0 and 12, respectively). The HDL_3_ particle size was increased slightly at Week 12 compared to the baseline, but no significance was detected. 

### 2.7. Change in HDL Quality during 12 Weeks

In HDL_2_, as indicated in Table 5, the extent of glycation (FI) and TG content in the particles was reduced by approximately −18% and −32%, respectively, while the particle diameter was increased significantly by approximately 6% at Week 12 compared to the baseline. In HDL_3_, the extent of glycation and TG content decreased by up to −16% and −9%, while the TC content enhanced significantly by approximately 32% compared to Week 0. These results suggest that the parameters of the HDL quality were improved, such as alleviated glycation and TG content with a rise in TC in both HDL_2_ and HDL_3_, by the policosanol intake and exercise. 

### 2.8. Change in apoA-I Expression in HDL

The male group demonstrated a remarkable enhancement in apolipoprotein (apo) A-I expression in HDL_2_ and HDL_3_ (up to 30% and 35% increase in band intensity, respectively) at Week 12 compared to Week 0 (baseline), suggesting that apoA-I was expressed more in HDL by policosanol intake and exercise (Figure 4). The band location of apoA-I (28 kDa) was shifted slightly up, as indicated by the red arrowhead with more multimerization at Week 0, suggesting that the molecular weight of apoA-I was augmented by glycation. On the other hand, the smeared and weaker band intensity at Week 0, indicated by the red arrowhead, was improved at Week 12 in both HDL_2_ and HDL_3_, resulting in a more distinct band intensity and smaller molecular weight. The female group also demonstrated a notable rise in apoA-I intensity within the HDL_2_ and HDL_3_ bands, representing increases of 35% and 48%, respectively, by the 12th week compared to the baseline at Week 0 (Appendix A). This suggests that the combination of policosanol intake and exercise led to the augmented expression of HDL-associated apoA-I.

### 2.9. Change in the Antioxidant Ability in HDL_2_ and HDL_3_

The HDL_2_-associated antioxidant abilities were elevated remarkably, showing an enhancement of up to 206% and 25% in the paraoxonase (PON) and ferric ion reduction ability (FRA), respectively, at Week 12 compared to the baseline (Figure 5A,B). The HDL_3_-associated PON and FRA improved by approximately 81% and 56%, respectively, at Week 12 compared to those of Week 0 (Figure 5C,D). These results suggest that policosanol intake and exercise increased the PON and FRA ability in HDL_2_ and HDL_3_. Interestingly, HDL_3_ displayed approximately 20-fold and 1.3-fold higher PON and FRA activities, respectively, than HDL_2_ at Weeks 0 and 12, suggesting that HDL_3_ exhibited a much higher antioxidant activity. 

### 2.10. Embryo Survivability under the Presence of Carboxymethyllysine 

The embryos injected with carboxymethyllysine (CML) alone exhibited the lowest survivability, approximately 35 ± 3% at 24 h post-injection. In contrast, embryos injected with phosphate-buffered saline (PBS) alone demonstrated the highest survivability (82 ± 3%) (Figure 6A). In the presence of CML, co-injection of HDL_3_ (10 ng of protein) in the male and female groups at Week 0 showed 47 ± 2% and 51 ± 3% survivability, respectively. Conversely, micro-injection of HDL_3_ in the male and female groups at Week 12 yielded a survivability of 70 ± 3% and 62 ± 1%, respectively. This suggests a significant enhancement in the antioxidant and anti-inflammatory activity of HDL_3_, effectively preventing embryo death from CML toxicity. 

The PBS-injected embryo exhibited the fastest development speed with a somite number of 30 ± 2. In contrast, the CML-injected embryo displayed the most attenuated developmental speed in eye pigmentation and tail elongation with the least somite number 25 ± 1 (Figure 6B). Under the presence of CML, the co-injection of HDL_3_ in the male and female groups at Week 0 resulted an attenuated developmental speed with a somite number of 26–27. On the other hand, the co-injection of HDL_3_ from the male and female groups at Week 12 led to a faster developmental speed with a somite number of 30–32, suggesting that the enhanced HDL_3_ functionality contributes to the faster developmental speed and higher survivability. 

Dihydroethidium (DHE) fluorescence staining revealed that the CML-alone injected embryo exhibited the highest reactive oxygen species (ROS) production (Figure 6C) (3.7-fold higher than PBS alone), suggesting that CML injection caused prompt ROS production via an acute inflammatory cascade. In contrast, the co-injection of HDL_3_ in either males or females from Week 0 displayed 53% or 27% lower ROS production, respectively, suggesting that the HDL exhibited adequate anti-inflammatory activity. A co-injection of HDL_3_ in males or females from Week 12 resulted in 80% or 87% lower ROS production, respectively, suggesting enhanced antioxidant and anti-inflammatory activity of HDL_3_. Acridine orange (AO) staining revealed that the CML-alone group exhibited a 2.8-fold greater extent of apoptosis compared to the PBS-alone group. However, co-injecting HDL_3_ in males or females from Week 0 resulted in an 18% reduction or 26% enhancement in apoptosis, respectively. This implies that the female HDL_3_ at Week 0 exhibited the worst quality regarding the extent of cellular apoptosis. On the other hand, the co-injection of HDL_3_ in males or females from Week 12 revealed a 39% or 60% lower extent of cellular apoptosis, respectively, suggesting the enhanced anti-apoptosis activity of HDL_3_.

## 3. Discussion

This study compares the effects of consuming policosanol along with high-intensity exercise over a 12-week period on the baseline values of BMI, serum lipid profile, and lipoprotein functionality in obese individuals who are following a regular diet without any supplementary interventions. Despite the conflicting data depending on age and gender, a meta-analysis with middle-aged and older adults, 45–64 years old, showed that aerobic and static exercise had a significant effect on improving only the SBP around the mean difference (MD) −9.3 and −10.5, respectively [11]. In contrast, aerobic and static exercise did not cause a significant reduction in the DBP [11]. In addition, a systematic review of metabolic syndrome in middle-aged women demonstrated that 12–24 weeks of exercise resulted in −0.57 kg, −0.43 kg/m^2^, −4.89 mmHg, and −2.71 mmHg changes in body weight, BMI, SBP, and DBP, respectively, which implies that exercise therapy by itself, whether involving aerobic or anaerobic activities, did not lead to a significant reduction in the DBP [27]. Overall, these results suggest that exercise therapy alone, regardless of aerobic and anaerobic, did not reduce the DBP significantly. 

On the contrary, the administration of Cuban policosanol (Raydel^®^, 20 mg) over a period of twelve weeks, without incorporating exercise, led to a notable reduction in both SBP and DBP among prehypertensive Korean subjects. Specifically, there was a decrease of −10.5 mmHg (−7.7% compared to the baseline) in SBP and −6.2 mmHg (−7.1% compared to the baseline) in DBP [20]. Interestingly, a lower dose of 10 mg of policosanol in the same study did not demonstrate a significant effect on blood pressure [20]. In a separate randomized trial spanning 12 weeks, normotensive Japanese participants experienced a reduction of −7.9 mmHg (−9.5% compared to the baseline) in SBP and −2.8 mmHg (−4.0% compared to the baseline) in DBP with policosanol intake [28]. Preceding these prior clinical investigations, we opted for a 20 mg dosage of policosanol in the current study, exploring policosanol effects in conjunction with exercise among obese patients. The Cuban policosanol consists of eight long-chain aliphatic alcohols ranging from 24 to 34 carbon atoms: 1-tetracosanol (C24, 0.1–20 mg/g); 1-hexacosanol (C26, 30.0–100.0 mg/g); 1-heptacosanol (C27, 1.0–30.0 mg/g); 1-octacosanol (C28, 600.0–700.0 mg/g); 1-nonacosanol (C29, 1.0–20.0 mg/g); 1-triacontanol (C30, 100.0–150.0); 1-dotriacontanol (C32, 50.0–100.0 mg/g); 1-tetratriacontanol (C34, 1.0–50.0 mg/g). Notably, Cuban policosanol is officially registered as a medicinal product in Central America and Latin America for the treatment of dyslipidemia and cardiovascular disease [29] via raising HDL-C and the improvement of HDL quality and functionality [20,21,22].

Meta-analysis with 19 studies documented that policosanol intake resulted in a larger reduction in the SBP (−3.4 mmHg) than the DBP (−1.5 mmHg) [30]. These results suggest that policosanol consumption was more likely to minimize the SBP than the DBP because exercise alone resulted in only a significant decrease in SBP. In the current results, however, a combination of Cuban policosanol intake and exercise resulted in a more dramatic decline in DBP (−10.6 mmHg) (*p* = 0.007, −12.8% from the baseline), whereas the decrease in SBP was approximately −9.8 mmHg (*p* = 0.046, −7.7% compared to the baseline). In particular, the male participants displayed a 3.2-fold-larger reduction in DBP (−16.7 mmHg, −19.4% compared to the baseline) than that of female participants (−5.1 mmHg, −6.4% from the baseline), as illustrated in Table 1. The results suggest a preliminary indication that the combination of policosanol intake and exercise may contribute to lowering both SBP and DBP. Nevertheless, a follow-up study incorporating a placebo control is necessary to confirm these findings and enable conclusive remarks.

The effects of exercise for 12 months on weight loss was an approximate 0–3% decrease from the baseline by aerobic and resistance training [31]. On the other hand, the current results showed a −12.5% body weight loss after 12 weeks of exercise and policosanol intake without calorie restriction. Although a sedentary lifestyle and lack of exercise are consistently linked with low HDL-C and high TG in obese subjects, whether exercise can elevate HDL-C quantity or improve HDL quality remains to be established. High-intensity exercise is associated with diminished body weight, BMI, total fat mass, and serum TG. Nevertheless, it is unclear if exercise can elevate the serum HDL-C. Exercise training has little effect on elevating the HDL-C levels in men with initially low HDL-C (<35 mg/dL) [32]. On the other hand, a meta-analysis with 25 articles indicated that aerobic exercise modestly raises the HDL-C level; approximately 2.5 mg/dL of HDL-C was increased by at least 900 kcal of energy expenditure or 120 min exercise per week [33]. The findings revealed a substantial increase in HDL-C/TC (%) and a decrease in TC, LDL-C, and TG/HDL-C (Table 2), implying a probable synergistic impact of daily intake of Cuban policosanol (20 mg) and regular exercise; nonetheless, it is essential to conduct a placebo-controlled study to validate the findings of the current research to establish the confirmatory remarks. The TG content in HDL_2_ was diminished, and the TC content in HDL_3_ was increased with the elevation of apoA-I in HDL_2_ and HDL_3_ at Week 12. 

The serum CoQ_10_ level was reduced by up to 40–50% by statin consumption due to the fundamental blocking of cholesterol biosynthesis through the inhibition of 3-hydroxy-3-methylglutaryl-CoA (HMG-CoA) reductase [34]. The depletion of CoQ_10_ was directly associated with statin-induced myopathy via mitochondrial dysfunction [35]. On the other hand, the current study showed that the 12 weeks of policosanol consumption did not interfere with the CoQ_10_ metabolism and homeostasis (approximately 370–513 ng/mL in serum), with a remarkable rise in CoQ_10_/LDL-C ratio (Figure 2A and Table 2), whereas the CoQ_10_/HDL-C ratio was not changed. These results suggest that although the LDL-C was curtailed by 20% at Week 12, the CoQ_10_ level was maintained sufficiently in the serum. Moreover, male participants exhibited a 38% boost in the serum CoQ_10_ level at Week 12. The larger increase in serum HDL-C was correlated with the higher rise in serum CoQ_10_ (Figure 1B).

It was postulated that statin medication depleted serum CoQ_10_ by blocking the synthesis of TC and LDL-C because LDL-C is a major carrier of CoQ_10_ in human blood. To the best of the authors’ knowledge, the current study is the first to document that Cuban policosanol (20 mg) intake for 12 weeks did not impair the serum CoQ_10_ metabolism despite remarkably lowering TC and LDL-C in participants of both genders. Furthermore, the enhancement in serum CoQ_10_ level was positively correlated with the increase in serum HDL-C (Figure 2B) without inhibiting HMG-CoA reductase. On the other hand, the decrease in LDL-C by atorvastatin 80 mg for 16 weeks of consumption was linked directly with a reduction in serum CoQ_10_ [36]. A larger portion of CoQ_10_ was carried by LDL (~58%) than by HDL (~26%) [37]. The current results present that the elevation of HDL-C by Cuban policosanol can be a strategy to maintain or elevate serum CoQ_10_ level because there has been no study to investigate an association between the increase in serum HDL-C and change in serum CoQ_10_. Further study will be needed to elucidate the association with CoQ_10_ in HDL subfractions to find the major carrier of CoQ_10_, either HDL_2_ or HDL_3_, upon the changes in cholesterol contents. 

Liver function enzymes, AST, ALT, and γ-GTP, were usually elevated by high-intensity exercise, particularly in weight training as an exercise-induced liver injury and rhabdomyolysis [24,38]. On the other hand, the current findings suggest a significant improvement in γ-GTP after 12 weeks, accompanied by decreases in AST and ALT, although the disparities observed are not statistically significant (Table 3). These findings suggest that the consumption of policosanol may mitigate liver damage against exercise-induced liver injury in obese individuals; however, a subsequent study incorporating a placebo control is imperative to validate and confirm the protective effect of policosanol against exercise-induced hepatic damage. Moreover, the extent of oxidation (MDA levels) in VLDL and LDL was reduced significantly (approximately −44% and −22%, respectively) (Table 4), whereas the antioxidant abilities in HDL_2_ and HDL_3_ were also elevated. All lipoprotein fractions, VLDL, LDL, HDL_2_, and HDL_3_, exhibited a reduction in the glycation extent of approximately −42%, −16%, −18%, and −16%, respectively, at Week 12 compared to the baseline (Table 4 and Table 5). Similarly, an ELSA-Brazil population study documented that non-alcoholic fatty liver disease (NAFLD) was associated with the elevation of the serum advanced glycation end products, as well as an increase in the serum TG, ALT, γ-GTP, and glycated hemoglobin with a decrease in HDL-C [39]. 

## 4. Materials and Methods

### 4.1. Policosanol 

The Raydel^®^ policosanol tablets (20 mg) were procured from Raydel Korea (Seoul, Republic of Korea). The tablets were produced using Cuban policosanol at Raydel Australia (Thornleigh, Sydney, Australia). The Cuban policosanol used in the manufacturing process was classified as authentic, with a specific ratio of each ingredient [40]. A comprehensive list of ingredients is provided in the Appendix A. 

### 4.2. Participants

Young and middle-aged obese volunteers (BMI > 28 kg/m^2^) were recruited randomly through a nationwide newspaper advertisement between September 2022 and December 2022. All participants voluntarily consented and provided informed consent. After recruiting, 20 volunteers (30–51 years old, BMI = 30.5 ± 1.1 kg/m^2^) participated in a daily exercise program, which was at least 120 min with high-intensity exercise per session combined with aerobic exercise for 60 min and weight training for 60 min. All participants were advised to consume the 20 mg policosanol tablet with a daily record of ingestion time. In addition, no nutrient supplements were allowed for all participants during the 12 weeks, because that may affect body weight, hepatic functions, and renal functions. The rationale for the selection of 20 mg policosanol was based on our previous findings, which demonstrated the effectiveness of the 20 mg dosage for 12 weeks in lowering both SBP and DBP in Korean hypertensive subjects [20] and normotensive Japanese participants [28].

The research received approval from the Korean National Institute for Bioethics Policy (KoNIBP) under the Korea Ministry of Health Care and Welfare (MOHW), with the approval number P01-202109-31-009. The study comprised 17 participants (n = 17), all of Korean ethnicity, including 8 males (n = 8) and 9 females (n = 9). The average age, BMI, and metabolic equivalent (MET) of the participants at Week 0 (baseline) are depicted in Table 6. For 12 weeks, the subjects adhered to a traditional Korean diet characterized by an abundance of proteins (15.1%), rice-derived carbohydrates (60.8%), and overall fat content (24.2%), comprising a variety of vegetables, meat, and fish. No caloric restrictions were imposed during the period. None of the participants consumed vegan or kosher diets. The exercise frequency, spending time, and intensity were estimated from a self-administered questionnaire inquiring about the frequency, time, and intensity of exercise per week during 12 weeks. The metabolic equivalents (METs) were determined utilizing a concise approach to estimate energy expenditure during physical activities, allowing for comparing intensity levels for distinct activities [41]. The METs score was computed from the participant’s survey responses and categorized as follows: light (<3.0 METs), moderate (3.0~6.0 METs), and vigorous (>6.0 METs). The total physical activity scores of participants during 12 weeks were calculated according to a “Guide to the assessment of physical activity” [42].

### 4.3. *Criteria of Inclusion and Exclusion*

The inclusion criteria for the participants were as follows: (1) age 25–55; (2) having a BMI > 28, but not exceeding BMI = 40; (3) individuals of apparently good health from both genders. The exclusion criteria were as follows: (1) lack of will and absence of motivation to finish the exercise program; (2) intense respiratory, cardiac, renal, hepatic, endocrinological, and metabolic disorders; (3) allergies; (4) excessive alcohol consumption, beyond 30 g of alcohol/day; (5) use of medications or functional food products, including statins and coenzyme Q_10_ supplements, which could impact lipid metabolism by either increasing HDL-C or reducing LDL-C levels, or lowering triglyceride levels; (6) taking nutrient supplements that may affect body weight, hepatic functions, and renal functions within three months prior to the study; (7) females who are pregnant, lactating, or intending to conceive during the research period; (8) individuals who have offered over 200 mL of blood within one month or 400 mL within three months prior to this program; (9) an individual who participated in another clinical trial within the last three months or was presently participating in another clinical study. 

### 4.4. Exercise Program

After recruiting and selection, all participants (n = 20) were split into four small groups (n = 5) and had to join a mandatory exercise program from Monday to Saturday at a public fitness center. The program, at least 120 min per day, consisted of aerobic exercise for at least 60 min with high intensity and weight training for at least 60 min with high endurance under supervision by a professional health trainer for each group. During the 12-week program, three participants (2 male and 1 female) quit because of their busy schedules and omitting policosanol intake. Seventeen participants completed the daily consumption of policosanol and the exercise program. 

### 4.5. Anthropometric Analysis

Certified technicians at the Seoul Eastern Branch of the Korea Health Care Association (located in Seoul, Republic of Korea) conducted measurements for all anthropometric and blood-pressure-related parameters, as outlined in Table 1. The height, body weight, body fat, and muscle weight were measured individually using INBODY770 (Inbody Co., Seoul, Republic of Korea). The blood pressure was measured using a digital automatic blood pressure monitor TM-2655P (A&D Co., Tokyo, Japan). 

### 4.6. Blood Analysis

Participants willingly offered the blood after a 12 h fasting time, following the Helsinki guidelines approved by the Institutional Review Board of the Korea National Institute for Bioethics Policy (KoNIBP, approval number P01-202109-31-009, approval date 27 September 2021), with support from the Ministry of Health Care and Welfare (MOHW) of the Republic of Korea. The serum lipid profiles (mentioned in Table 2), and protein parameters (mentioned in Table 3) were quantified using an automatic analyzer (Cobas C502 chemistry analyzer, Roche, Germany) employing SCL Healthcare’s (Seoul, Republic of Korea) commercial diagnostic service. 

### 4.7. Quantification of Serum Coenzyme Q_10_

The CoQ_10_ concentration in serum was measured utilizing the CUSABIO human CoQ_10_ enzyme-linked immunosorbent assay (ELISA) kit (Cat# CSB-E14081h, Cusabio Biotechnology Inc. Houston, TX, USA) as the suggested guidelines recommended by the manufacturer. For CoQ_10_ quantification, a 100-fold diluted serum from each subject was employed. 

### 4.8. Isolation of Lipoproteins

Very-low-density lipoproteins (VLDL, d < 1.019), LDL (1.019 < d < 1.063), HDL_2_ (1.063 < d < 1.125), and HDL_3_ (1.125 < d < 1.225) were separated from the respective serum via sequential ultracentrifugation with the density regulated by supplementing NaCl and NaBr following the usual procedures [43]. Briefly, each serum with the modified density was ultracentrifuged sequentially at 100,000× *g* for 24 h at 10 °C utilizing a Himac NX (Hitachi, Tokyo, Japan) furnished with a fixed angle rotor P50AT4-0124. The isolated lipoproteins were individually gathered and subjected to dialysis to eliminate residual NaCl and NaBr using Tris-buffered saline (TBS; 10 mM Tris-HCl, 140 mM NaCl, and 5 mM ethylene-diamine-tetraacetic acid (EDTA) [pH 8.0]).

### 4.9. Characterization of Lipoproteins

Individual lipoproteins, VLDL, LDL, HDL_2_, and HDL_3_, were characterized to determine the lipid and protein composition, oxidation extent, and glycation extent. For each isolated lipoprotein fraction, we utilized commercially available kits (Cleantech TS-S; Wako Pure Chemical, Osaka, Japan) to measure total cholesterol (TC) and triglyceride (TG) levels. The protein concentrations of the lipoproteins were assessed through a Lowry protein assay, as adapted by Markwell et al. [44], employing Folin–Ciocalteu’s phenol reagent (F9252, Sigma–Aldrich, St. Louis, MO, USA) with bovine serum albumin (BSA) as a reference.

At an equivalent protein concentration within each lipoprotein, the assessment of individual lipoprotein oxidation was conducted by quantifying the concentration of oxidized species through the thiobarbituric acid reactive substances (TBARS) procedure, employing malondialdehyde (MDA) as a reference [45]. The level of glycation in individual lipoproteins was determined by measuring fluorometric intensity at 370 nm (excitation) and 440 nm (emission) under identical protein concentrations, as previously outlined [46]. This analysis was performed using FL6500 spectrofluorometer (Perkin-Elmer, Norwalk, CT, USA).

The expression of apolipoproteins in HDL_2_ (2 mg of protein/mL) and HDL_3_ (2 mg of protein/mL) were compared using 15% SDS-PAGE under a denatured state. The visualization of proteins band was achieved using 0.125% Coomassie brilliant blue. Subsequently, the relative intensities of the bands between Weeks 0 and 12 were compared through band scanning, employing Gel Doc^®^ XR (Bio-Rad, Hercules, CA, USA) along with Quantity One software (version 4.5.2).

### 4.10. Transmitted Electron Microscopy 

Transmitted electron microscopy (TEM) was performed at an acceleration voltage of 80 kV using a Hitachi H-7800 (Ibaraki, Japan) at Raydel Research Institute (Daegu, Republic of Korea). Each lipoprotein underwent negative staining using 1.5% sodium phosphotungstate (PTA) with pH 7.4, and a final apolipoprotein concentration of 0.2 mg/mL in TBS. A 5 μL aliquot of the lipoprotein suspension was blotted with filter paper, immediately replaced with a 5 μL droplet of the PTA. Following a brief period, the stained HDL fraction was blotted onto a Formvar carbon-coated 300 mesh copper grid and allowed to air-dry. The morphology of each lipoprotein was determined by TEM at 40,000× magnification, using EMIP-EX software, Version 07.13 (Hitachi, Tokyo, Japan), in accordance with a previously documented protocol [28].

### 4.11. Agarose Gel Electrophoresis 

The participants’ sample electromobility was assessed by analyzing LDL migration using agarose gel electrophoresis [47]. A 0.5% agarose gel (120 mm length × 60 mm width × 5 mm thickness) was prepared, and electrophoresis was carried out with 50 V for 1 h in Tris-acetate-ethylene-diamine-tetraacetic acid buffer (pH 8.0). The relative electrophoretic mobility depends on the intact charge and three-dimensional structure of LDL. The gels were dried and stained with 1.25% Coomassie brilliant blue, after which the relative band intensities were compared by band scanning using Gel Doc^®^ XR (Bio-Rad) with Quantity One software (version 4.5.2).

### 4.12. Antioxidant Activities in the HDL

The evaluation of paraoxonase-1 (PON-1) activity towards paraoxon involved assessing the enzymatic hydrolysis of paraoxon into *p*-nitrophenol and diethylphosphate. This process was catalyzed by the enzyme [48]. To conduct this analysis, 20 μL of equally diluted HDL at 1 mg/mL was combined with 180 μL of paraoxon-ethyl (Sigma Cat. No. D-9286) in the presence of a buffer solution (90 mM Tris-HCl/3.6 mM NaCl/2 mM CaCl_2_ [pH 8.5]). The determination of the PON-1 activity involved measuring the initial velocity of *p*-nitrophenol production at 37 °C, as estimated by the absorbance at 415 nm (microplate reader, Bio-Rad model 680; Bio-Rad, Hercules, CA, USA).

The ferric ion-reducing ability (FRA) was determined using the method reported by Benzie and Strain [49]. Detailed procedures are available in the Appendix A. The antioxidant capabilities of individual HDL samples were assessed by quantifying the rise in absorbance resulting from the generation of ferrous ions. For this analysis, each HDL (100 μg protein/mL) was combined with 300 μL of freshly prepared FRA reagent to serve as an antioxidant source.

### 4.13. Zebrafish Maintenance

Zebrafish and their embryos were maintained using standard protocols [50] and in compliance with the Guide for the Care and Use of Laboratory Animals [51]. All zebrafish-related procedures and maintenance were approved by the Committee of Animal Care and Use of Raydel Research Institute (approval code RRI-20-003, approval date 3 January 2020, Daegu, Republic of Korea). The detailed procedure is available in the Appendix A.

### 4.14. Microinjection of CML and HDL into Zebrafish Embryos

At one day post-fertilization (dpf), individual zebrafish embryos underwent microinjection using a pneumatic picopump (PV830; World Precision Instruments, Sarasota, FL, USA) provided with a magnetic manipulator (MM33; Kantec, Bensenville, IL, USA) and microcapillary pipette-pulling device (PC-10; Narishigen, Tokyo, Japan). The injection involved delivering each HDL_3_ (10 ng protein) or co-injecting with 500 ng CML at the same yolk location to minimize bias, employing a method previously described [52]. Following the injection, a stereomicroscope (Zeiss Stemi 305, Oberkochen, Germany) was used to observe the live embryos and photographed at 20× magnification using a ZEISS Axiocam 208 color (Jena, Germany). The chorion was removed 24 h post-injection, and each live embryo was compared to assess the developmental stage at a higher magnification of 50×.

### 4.15. Imaging of Oxidative Stress, Apoptosis in Embryo

Following the injection of CML with individual HDL, the level of reactive oxygen species (ROS) and the extent of cellular apoptosis in the embryos were determined by dihydroethidium (DHE) staining and acridine orange (AO) staining, respectively, as described elsewhere [53]. Fluorescence observations (excitation = 585 nm and emission = 615 nm) were utilized to capture images of ROS, as outlined previously [54]. The comparison of cellular apoptosis across the groups was conducted through acridine orange (AO) staining and fluorescence observations (excitation = 505 nm, emission = 535 nm), as in earlier described methods [55]. 

### 4.16. Statistical Analysis

The findings are reported as the mean ± SEM, derived from at least three independent experiments with duplicate samples. All analyses underwent normalization via a homogeneity test of variance using Levene’s statistics. In cases where normalization was not achieved, nonparametric statistics, specifically the Kruskal–Wallis test, were employed. A paired *t*-test was used to assess the statistical significance between the baseline and follow-up values within the groups. Spearman correlation analysis was performed to identify the positive or negative associations between the serum level of CoQ_10_ and the lipid profile and blood pressure. All statistical analyses were conducted using the SPSS software package version 29.0 (SPSS Inc., Chicago, IL, USA).

## 5. Limitations and Perspectives

The basic limitation of the study emerges with the small number of participants. Furthermore, including the control (placebo and exercises) group would be of great interest to compare the findings with the policosanol (PCO) group. The present study was the preliminary effort that laid down a foundation for future research employing a large pool of participants. In the prospective study, a large pool of participants will be distributed into four distinct groups, i.e., PCO or placebo-consuming groups without exercise, while the other groups will be PCO or placebo consumption along with extensive exercise. Also, the phytochemicals as minor constituents and their quantities present in the Cuban policosanol will be determined and compared with the policosanols from other sources to assess the specific influence of these constituents on functionality, particularly in relation to hypertension and dyslipidemia. The outcome will provide conclusive insights into the policosanols from various sources on obesity, hypertension, dyslipidemia, lipoprotein quality, functionality, and the impact on the serum coenzyme Q_10_.

## 6. Conclusions

As a preliminary study, the findings from this research provide basic insights suggesting that the combination of high-intensity exercise and Cuban policosanol intake in obese subjects improve hypertension and dyslipidemia by enhancing HDL quality and antioxidant functionality without destabilizing the CoQ_10_ metabolism or causing liver damage. In HDL_2_ and HDL_3_, an enlarged particle size and increased particle numbers, along with elevated apoA-I content and antioxidant activities, were observed after 12 weeks of therapy. The HDL derived from the group engaged in exercise and consumption of Cuban policosanol exhibited a significant protective influence on zebrafish embryos against ROS-mediated apoptosis induced by CML, highlighting the beneficial effects of policosanol on the functionality of HDL. These results could serve as a basis for future large clinical trials with obese participants in placebo-controlled, randomized, and double-blinded studies.

## Figures and Tables

**Figure 1 pharmaceuticals-17-00132-f001:**
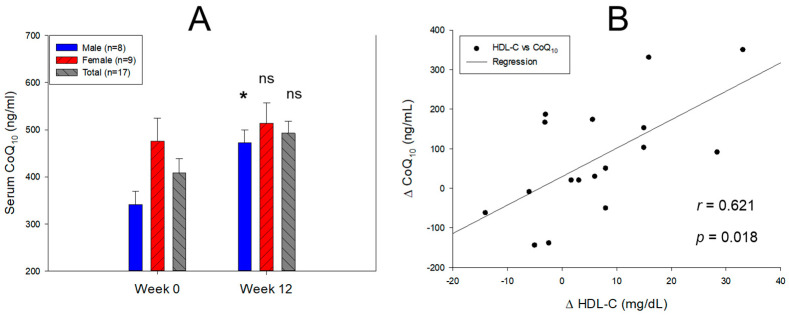
Comparison of the serum coenzyme Q_10_ (CoQ_10_) level at Week 12 from the baseline (Week 0). (**A**). Quantification of CoQ_10_ in 100-fold diluted serum from each participant using Cusabio human CoQ_10_ ELİSA kit. *, *p* < 0.05 versus Week 0; ns, not significant. (**B**). Spearman correlation analysis with the net change in HDL-C (ΔHDL-C) and net change in CoQ_10_ (ΔCoQ10) between Weeks 0 and 12.

**Figure 2 pharmaceuticals-17-00132-f002:**
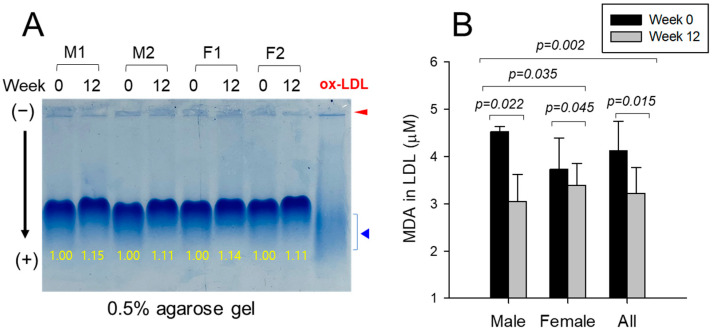
Comparison of the band intensity and electromobility depends on oxidation extent in LDL between Weeks 0 and 12. (**A**) Electrophoresis under the nondenatured state on agarose gel (final 0.5%) with 50 V for 1 h. The apo-B in LDL was visualized by Coomassie brilliant blue staining (final 1.25%). The yellow font indicates the band intensity compared to the 100% initialized band intensity of Week 0. Oxidized LDL, cupric ion (final 10 μM) treated for 4 h. The red arrowhead indicates an aggregated oxLDL band at the loading position. The blue triangle indicates the smeared and disappeared oxLDL band range. M1, M2, F1, and F2 are representative band images of the participants. M, male; F, female. (**B**) Determination of the oxidized species amount in LDL by a thiobarbituric acid reactive substance assay using a malondialdehyde (MDA) standard. The data are expressed as the mean ± SEM from three independent experiments with duplicate samples. The oxidation extent in each group between Weeks 0 and 12 was compared using a paired *t*-test.

**Figure 3 pharmaceuticals-17-00132-f003:**
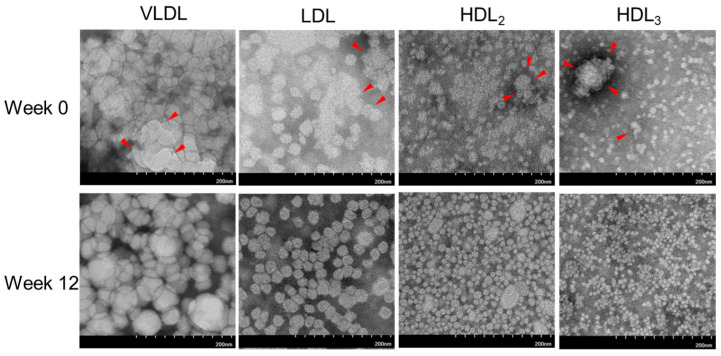
Transmitted electron microscopy (TEM) image analysis of lipoproteins, VLDL, LDL, HDL_2_, and HDL_3_, and size analysis between Weeks 0 and 12. Representative image of VLDL, LDL, HDL_2_, and HDL_3_ from the same individual between Weeks 0 and 12 with a magnification of 150,000×. One graduation of the scale bar indicates 20 nm. The red arrowhead indicates aggregated lipoprotein particles. (**A**) Comparison of the VLDL particle size in each group between Week 0 and 12 using a paired *t*-test. ns, not significant versus Week 0. (**B**) Comparison of the LDL particle size in each group between Weeks 0 and 12 using a paired *t*-test. ns, not significant versus Week 0. (**C**) Comparison of the HDL_2_ particle size in each group between Weeks 0 and 12 using a paired *t*-test. *, *p* < 0.05 versus Week 0. (**D**) Observation of the HDL_3_ particle size in each group between Weeks 0 and 12 using a paired *t*-test. ns, not significant versus Week 0.

**Figure 4 pharmaceuticals-17-00132-f004:**
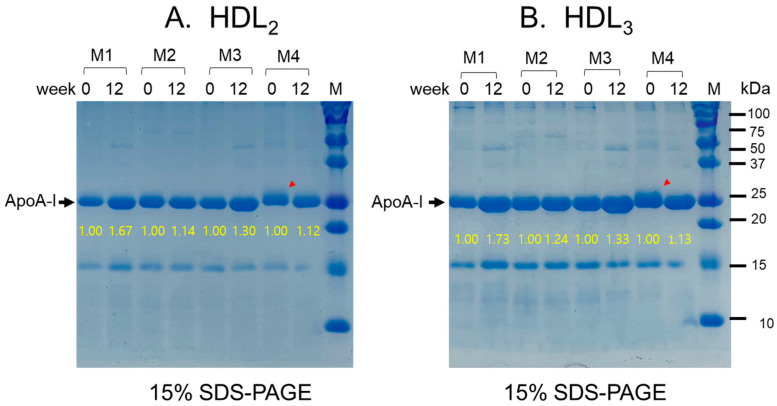
Representative image of the apoA-I expression pattern in HDL_2_ (**A**) and HDL_3_ (**B**) between Weeks 0 and 12 in the male group. M1, M2, M3, and M4 represent Male Participants 1, 2, 3, and 4, respectively, as a representative image. The yellow font indicates the band intensity of apoA-I compared to Week 0. The red arrowhead indicates a smeared band intensity with a shifted-up band position of apoA-I because of glycation at Week 0.

**Figure 5 pharmaceuticals-17-00132-f005:**
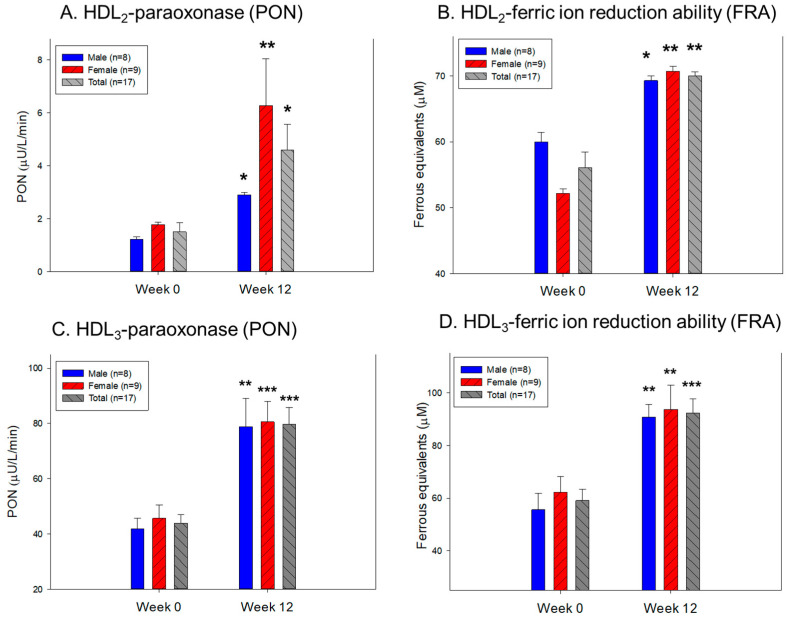
Determination of paraoxonase (PON)-1 activity and ferric ion reduction ability (FRA) in HDL_2_ and HDL_3_ at Week 0 and Week 12. *, *p* < 0.05 versus Week 0; **, *p* < 0.01 versus Week 0; ***, *p* < 0.001 versus Week 0. (**A**). Comparison of HDL_2_-associated PON-1 activity at Weeks 0 and 12. (**B**). Comparison of HDL_2_-associated FRA activity at Weeks 0 and 12. (**C**). Comparison of HDL_3_-associated PON-1 activity at Weeks 0 and 12. (**D**). Comparison of HDL_3_-associated FRA activity at Weeks 0 and 12.

**Figure 6 pharmaceuticals-17-00132-f006:**
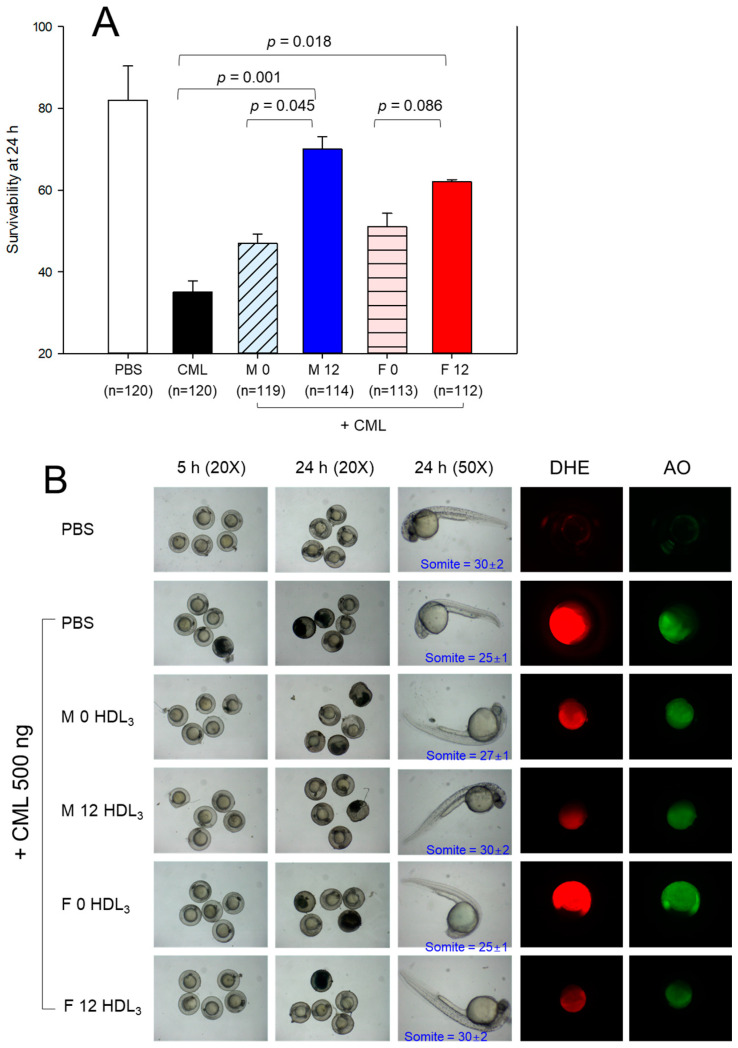
Comparison of the survivability, developmental speed and morphology, and extent of ROS production and cellular apoptosis after injection of HDL_3_ in each group under the presence of carboxymethyllysine. (**A**). Survivability of embryo during 24 h post-injection. (**B**). Morphological change in an embryo during development. Stereo image observation at 5 h and 24 h post-injection, somite number at 24 h post-injection, DHE-stained image at 5 h post-injection, AO-stained image at 5 h post-injection. (**C**). Quantification of ROS production and extent of cellular apoptosis from DHE staining and AO staining using Image J software version 1.53 (http://rsb.info.nih.gov/ij/, accessed on 16 May 2023). The data are expressed as the mean ± SEM from three independent experiments. **, *p* < 0.01 versus CML alone in DHE; ***, *p* < 0.01 versus CML alone in DHE; ^#^, *p* < 0.05 versus CML alone in AO; ^###^, *p* < 0.001 versus CML alone in AO.

**Table 1 pharmaceuticals-17-00132-t001:** Change in anthropometric profiles between Week 0 and Week 12.

	Groups	Week 0	Week 12	Δ Change (%)	*p* ^†^
		Mean ± SEM	Mean ± SEM		
SBP (mmHg)	Male (n = 8)	133.8 ± 6.8	119.1 ± 5.2	−10.9	0.109
	Female (n = 9)	122.8 ± 2.8	117.2 ± 3.5	−4.5	0.233
	Total (n = 17)	127.9 ± 3.7	118.1 ± 3.0	−7.7	0.046
DBP (mmHg)	Male (n = 8)	85.8 ± 5.2	69.1 ± 3.9	−19.4	0.022
	Female (n = 9)	79.3 ± 2.1	74.2 ± 3.0	−6.4	0.180
	Total (n = 17)	82.4 ± 2.7	71.8 ± 2.4	−12.8	0.007
BMI (kg/m^2^)	Male (n = 8)	31.0 ± 1.9	25.0 ± 2.7	−19.3	0.092
	Female (n = 9)	29.0 ± 1.4	26.0 ± 1.4	−10.5	0.146
	Total (n = 17)	30.0 ± 1.1	25.5 ± 1.4	−14.8	0.022
Weight (kg)	Male (n = 8)	99.9 ± 6.4	85.2 ± 5.5	−14.7	0.104
	Female (n = 9)	79.0 ± 4.1	71.0 ± 4.2	−10.2	0.190
	Total (n = 17)	88.8 ± 4.4	77.7 ± 3.7	−12.5	0.064
Waist circumference (cm)	Male (n = 7)	108.3 ± 3.8	89.7 ± 4.6	−17.2	0.009
	Female (n = 9)	95.8 ± 3.7	87.4 ± 4.9	−8.8	0.193
	Total (n = 16)	101.2 ± 3.1	88.4 ± 3.3	−12.7	0.008
Muscle mass (kg)	Male (n = 8)	65.3 ± 3.2	64.1 ± 2.8	−1.8	0.786
	Female (n = 9)	45.2 ± 1.4	44.2 ± 1.4	−2.2	0.625
	Total (n = 17)	54.6 ± 3.0	53.6 ± 2.9	−2.0	0.797
Total fat mass (kg)	Male (n = 8)	30.6 ± 3.4	17.1 ± 3.4	−44.0	0.014
	Female (n = 9)	31.0 ± 3.4	24.0 ± 3.4	−22.6	0.163
	Total (n = 17)	30.8 ± 2.3	20.8 ± 2.5	−32.6	0.006
Subcutaneous fat mass (kg)	Male (n = 8)	28.9 ± 3.4	15.8 ± 3.4	−45.4	0.017
	Female (n = 9)	29.7 ± 3.3	22.9 ± 3.3	−23.1	0.162
	Total (n = 17)	29.3 ± 2.3	19.5 ± 2.5	−33.4	0.007
Visceral fat mass (kg)	Male (n = 8)	1.7 ± 0.2	1.4 ± 0.2	−20.4	0.229
	Female (n = 9)	1.3 ± 0.1	1.2 ± 0.1	−11.8	0.200
	Total (n = 17)	1.5 ± 0.1	1.3 ± 0.1	−16.4	0.109
Body fat percentage (%)	Male (n = 8)	30.1 ± 1.6	19.2 ± 2.8	−36.3	0.005
	Female (n = 9)	38.4 ± 2.8	32.7 ± 3.1	−14.9	0.189
	Total (n = 17)	34.5 ± 1.9	26.3 ± 2.7	−23.7	0.018
Body water content (kg)	Male (n = 8)	50.8 ± 2.5	50.0 ± 2.2	−1.6	0.803
	Female (n = 9)	35.2 ± 1.1	34.4 ± 1.1	−2.2	0.626
	Total (n = 17)	42.5 ± 2.3	41.7 ± 2.2	−1.9	0.806

Data are presented as the mean ± SEM (standard error of the mean). *p*^†^, paired *t*-test performed for values obtained between Weeks 0 and 12. SBP, systolic blood pressure; DBP, diastolic blood pressure; BMI, body mass index.

**Table 2 pharmaceuticals-17-00132-t002:** Change in the serum lipid parameters and coenzyme Q_10_ (CoQ_10_) ratio between Weeks 0 and 12.

	Groups	Week 0	Week 12	Δ Change (%)	*p* ^†^
		Mean ± SEM	Mean ± SEM		
TC (mg/dL)	Male (n = 8)	267.7 ± 24.2	211.2 ± 12.5	−21.1	0.056
	Female (n = 9)	221.4 ± 8.2	195.3 ± 15.3	−11.8	0.152
	Total (n = 17)	243.2 ± 13.1	202.8 ± 9.9	−16.6	0.019
TG (mg/dL)	Male (n = 8)	191.4 ± 34.8	84.1 ± 17.4	−56.1	0.015
	Female (n = 9)	102.2 ± 9.5	52.4 ± 6.5	−48.7	0.001
	Total (n = 17)	144.2 ± 19.9	67.3 ± 9.4	−53.3	0.002
RC (mg/dL)	Male (n = 8)	36.2 ± 7.7	16.2 ± 3.4	−55.4	0.032
	Female (n = 9)	20.5 ± 1.9	12.7 ± 2.2	−38.0	0.017
	Total (n = 17)	27.9 ± 4.1	14.3. ± 2.0	−48.6	0.007
HDL-C (mg/dL)	Male (n = 8)	43.8 ± 3.1	54.3 ± 4.7	23.9	0.084
	Female (n = 9)	56.3 ± 3.5	58.8 ± 5.9	4.4	0.721
	Total (n = 17)	50.4 ± 2.8	56.6 ± 3.7	12.4	0.189
HDL-C/TC (%)	Male (n = 8)	17.1 ± 1.7	26.2 ± 2.4	53.2	0.008
	Female (n = 9)	25.7 ± 1.9	30.5 ± 2.3	18.6	0.126
	Total (n = 17)	21.6 ± 1.6	28.5 ± 1.7	31.5	0.007
TG/HDL-C (ratio)	Male (n = 8)	4.7 ± 0.9	1.7 ± 0.4	−63.2	0.014
	Female (n = 9)	1.9 ± 0.2	1.0 ± 0.1	−48.6	0.005
	Total (n = 17)	3.2 ± 0.5	1.3 ± 0.2	−58.6	0.005
LDL-C (mg/dL)	Male (n = 8)	188.0 ± 18.7	140.8 ± 10.3	−25.1	0.044
	Female (n = 9)	144.6 ± 8.3	123.7 ± 11.7	−14.5	0.164
	Total (n = 17)	165.0 ± 10.9	131.7 ± 7.9	−20.2	0.019
LDL-C/HDL-C (ratio)	Male (n = 8)	4.5 ± 0.6	2.8 ± 0.4	−38.2	0.029
	Female (n = 9)	2.7 ± 0.3	2.2 ± 0.2	−17.9	0.198
	Total (n = 17)	3.5 ± 0.4	2.5 ± 0.2	−30.0	0.022
Serum CoQ_10_ (μmol/L)	Male (n = 8)	0.395 ± 0.032	0.547 ± 0.031	38.2	0.027
	Female (n = 9)	0.551 ± 0.056	0.595 ± 0.05	7.9	0.566
	Total (n = 17)	0.473 ± 0.035	0.571 ± 0.029	20.6	0.090
CoQ_10_/TC (μmol/mol)	Male (n = 8)	59.4 ± 7.3	102.5 ± 11.1	72.6	0.007
	Female (n = 9)	99.5 ± 17.0	127.3 ± 20.9	27.9	0.323
	Total (n = 17)	79.5 ± 10.5	114.9 ± 11.9	44.5	0.034
CoQ_10_/HDL-C (μmol/mol)	Male (n = 8)	369.7 ± 75.0	395.8 ± 39.7	7.1	0.764
	Female (n = 9)	400.5 ± 92.2	415.5 ± 75.3	3.7	0.902
	Total (n = 17)	385.1 ± 57.3	405.6 ± 41.0	5.3	0.773
CoQ_10_/LDL-C (μmol/mol)	Male (n = 8)	85.9 ± 12.5	156.3 ± 19.4	82.0	0.010
	Female (n = 9)	155.1 ± 25.1	207.5 ± 34.2	33.8	0.240
	Total (n = 17)	120.5 ± 16.6	181.9 ± 20.2	51.0	0.026

The data are presented as the mean ± SEM (standard error of the mean). *p*^†^, paired *t*-test performed for the values obtained between Weeks 0 and 12. CoQ_10_, coenzyme Q_10_; TC, total cholesterol; TG, triglyceride, RC, remnant cholesterol; HDL-C, cholesterol content in high-density lipoprotein; LDL-C, cholesterol content in low-density lipoprotein.

**Table 3 pharmaceuticals-17-00132-t003:** Change in the serum protein parameters between Weeks 0 and 12.

	Groups	Week 0	Week 12	Δ Change (%)	*p* ^†^
		Mean ± SEM	Mean ± SEM		
AST (IU/L)	Male (n = 8)	33.5 ± 5.1	27.5 ± 2.2	−17.9	0.306
	Female (n = 9)	23.7 ± 2.8	27.4 ± 2.8	16.0	0.352
	Total (n = 17)	28.3 ± 3.0	27.5 ± 1.7	−2.9	0.813
ALT (IU/L)	Male (n = 8)	37.8 ± 6.4	25.3 ± 2.8	−33.1	0.103
	Female (n = 9)	29.2 ± 8.3	28.6 ± 7.0	−2.3	0.952
	Total (n = 17)	33.2 ± 5.3	27.0 ± 3.8	−18.8	0.346
γ-GTP (IU/L)	Male (n = 8)	48.0 ± 11.6	19.9 ± 2.4	−58.6	0.046
	Female (n = 9)	30.4 ± 6.9	15.6 ± 1.7	−48.9	0.065
	Total (n = 17)	38.7 ± 6.7	17.6 ± 1.5	−54.6	0.007
hsCRP (mg/L)	Male (n = 8)	2.7 ± 1.1	3.4 ± 2.3	28.5	0.772
	Female (n = 9)	2.0 ± 0.5	2.0 ± 0.7	−0.6	0.989
	Total (n = 17)	2.3 ± 0.6	2.7 ± 1.1	15.2	0.781
apoA-I (mg/dL)	Male (n = 8)	144.5 ± 13.6	132.6 ± 8.7	−8.2	0.473
	Female (n = 9)	140.9 ± 7.8	154.8 ± 10.4	9.9	0.303
	Total (n = 17)	142.6 ± 7.4	144.4 ± 7.2	1.2	0.865
apo-B (mg/dL)	Male (n = 8)	140.8 ± 16.9	105.0 ± 7.2	−25.4	0.072
	Female (n = 9)	89.9 ± 6.3	95.1 ± 7.3	5.8	0.596
	Total (n = 17)	113.8 ± 10.5	99.8 ± 5.1	−12.4	0.237
apo-B/apoA-I	Male (n = 8)	1.0 ± 0.1	0.8 ± 0.1	−19.9	0.195
	Female (n = 9)	0.7 ± 0.1	0.6 ± 0.1	−4.6	0.793
	Total (n = 17)	0.8 ± 0.1	0.7 ± 0.0	−13.4	0.277
Glucose (mg/dL)	Male (n = 8)	105.1 ± 5.7	97.3 ± 5.0	−7.5	0.320
	Female (n = 9)	90.3 ± 3.7	89.2 ± 4.4	−1.2	0.849
	Total (n = 17)	97.3 ± 3.7	93.0 ± 3.4	−4.4	0.398
Creatinine (mg/dL)	Male (n = 8)	1.1 ± 0.0	1.1 ± 0.1	−3.0	0.619
	Female (n = 9)	1.0 ± 0.1	1.0 ± 0.0	−2.2	0.804
	Total (n = 17)	1.1 ± 0.0	1.0 ± 0.0	−2.6	0.627
e-GRF (mL/min/1.73 m^2^)	Male (n = 8)	76.7 ± 2.9	80.0 ± 3.8	4.3	0.505
	Female (n = 9)	72.6 ± 4.2	70.1 ± 4.3	−3.4	0.689
	Total (n = 17)	74.4 ± 2.7	74.4 ± 3.1	0.1	0.988

The data are displayed as the mean ± SEM (standard error of the mean). *p*^†^, paired *t*-test performed for values obtained between Weeks 0 and 12. hs-CRP, high sensitivity C-reactive protein; AST, aspartate transferase; ALT, alanine aminotransferase; *γ*-GTP, gamma-glutamyl transferase; e-GRF, estimated glomerular filtration rate.

**Table 4 pharmaceuticals-17-00132-t004:** Lipid compositions and extent of VLDL and LDL between Weeks 0 and 12.

		Groups	Week 0	Week 12	Δ Change (%)	*p* ^†^
			Mean ± SEM	Mean ± SEM		
VLDL	FI (Glycated)	Male (n = 8)	7131 ± 910	4431 ± 374	−37.9	0.052
		Female (n = 9)	9311 ± 2943	5085 ± 782	−45.4	0.214
		All (n = 17)	8221 ± 1484	4758 ± 420	−42.1	0.041
	MDA (μM)	Male (n = 8)	27.1 ± 4.8	15.0 ± 1.9	−44.6	0.080
		Female (n = 9)	18.0 ± 6.5	10.4 ± 3.4	−42.4	0.339
		All (n = 17)	22.6 ± 4.1	12.7 ± 2.0	−43.7	0.050
	Diameter (nm)	Male (n = 8)	37.6 ± 0.4	39.4 ± 2.8	4.8	0.588
		Female (n = 9)	38.8 ± 1.0	38.5 ± 2.6	−0.8	0.918
		Total (n = 17)	38.1 ± 0.5	38.9 ± 1.7	2.2	0.659
	TC (μg/mg of protein)	Male (n = 8)	59.3 ± 10.1	53.1 ± 4.5	−10.5	0.582
		Female (n = 9)	66.6 ± 7.3	41.2 ± 3.4	−38.2	0.009
		Total (n = 17)	63.2 ± 6.0	46.8 ± 3.1	−26.0	0.023
	TG (μg/mg of protein)	Male (n = 8)	120.6 ± 19.4	74.0 ± 15.3	−38.6	0.081
		Female (n = 9)	131.4 ± 16.4	68.4 ± 10.8	−47.9	0.006
		Total (n = 17)	126.3 ± 12.3	71.0 ± 8.9	−43.7	0.001
LDL	FI (Glycated)	Male (n = 8)	5009 ± 241	4358 ± 143	−13.0	0.040
		Female (n = 9)	4907 ± 248	4138 ± 165	−15.7	0.020
		Total (n = 17)	4955 ± 168	4242 ± 110	−14.4	0.001
	MDA (μM)	Male (n = 8)	4.5 ± 0.1	3.0 ± 0.3	−32.7	0.019
		Female (n = 9)	3.7 ± 0.4	3.4 ± 0.3	−9.1	0.494
		All (n = 17)	4.1 ± 0.2	3.2 ± 0.2	−22.0	0.012
	Diameter (nm)	Male (n = 8)	25.8 ± 0.7	27.4 ± 0.4	5.9	0.091
		Female (n = 9)	26.7 ± 0.8	25.8 ± 0.6	−3.5	0.367
		Total (n = 17)	26.3 ± 0.5	26.5 ± 0.4	0.9	0.740
	TC (μg/mg of protein)	Male (n = 8)	139.8 ± 14.6	103.2 ± 5.0	−26.2	0.043
		Female (n = 9)	150.1 ± 19.4	95.3 ± 4.7	−36.5	0.023
		Total (n = 17)	145.3 ± 12.1	99.0 ± 3.5	−31.8	0.002
	TG (μg/mg of protein)	Male (n = 8)	20.6 ± 2.1	12.2 ± 1.3	−40.8	0.005
		Female (n = 9)	19.7 ± 2.6	10.8 ± 1.7	−45.2	0.010
		Total (n = 17)	20.1 ± 1.6	11.4 ± 1.1	−43.1	<0.001

Data are presented as the mean ± SEM (standard error of the mean). *p*^†^, paired *t*-test performed for values obtained between Weeks 0 and 12. VLDL, very-low-density lipoprotein; FI, fluorescence intensity (excitation = 370 nm, emission = 440 nm, 0.01 mg/dL of protein); MDA, malondialdehyde; TC, total cholesterol (μg/mg of protein); TG, triglyceride (μg/mg of protein); LDL, low-density lipoprotein.

**Table 5 pharmaceuticals-17-00132-t005:** Parameters of the HDL quality and functionality. Glycation and oxidation extent, lipid compositions, paraoxonase activity, and ferric ion reduction ability in HDL particles between Weeks 0 and 12.

		Groups	Week 0	Week 12	Δ Change %	*p* ^†^
			Mean ± SEM	Mean ± SEM		
HDL_2_	FI (Glycated)	Male (n = 8)	2234 ± 213	1837 ± 180	−17.8	0.175
		Female (n = 9)	1969 ± 166	1616 ± 153	−17.9	0.138
		Total (n = 17)	2094 ± 133	1720 ± 117	−17.9	0.043
	Diameter (nm)	Male (n = 8)	12.8 ± 0.4	13.5 ± 0.2	5.6	0.112
		Female (n = 9)	12.5 ± 0.3	13.2 ± 0.4	6.0	0.180
		Total (n = 17)	12.6 ± 0.2	13.4 ± 0.2	5.8	0.038
	TC (μg/mg of protein)	Male (n = 8)	84.7 ± 12.5	67.2 ± 4.8	−20.6	0.225
		Female (n = 9)	68.7 ± 7.7	69.7 ± 3.6	1.4	0.910
		Total (n = 17)	76.2 ± 7.2	68.5 ± 2.9	−10.1	0.332
	TG (μg/mg of protein)	Male (n = 8)	14.9 ± 2.8	7.5 ± 1.2	−49.4	0.029
		Female (n = 9)	9.8 ± 0.9	6.7 ± 1.0	−32.2	0.033
		Total (n = 17)	12.2 ± 1.5	7.1 ± 0.8	−42.1	0.004
HDL_3_	FI (Glycated)	Male (n = 8)	1885 ± 208	1616 ± 159	−14.3	0.322
		Female (n = 9)	1891 ± 241	1578 ± 136	−16.6	0.278
		Total (n = 17)	1888 ± 156	1596 ± 101	−15.5	0.126
	Diameter (nm)	Male (n = 8)	9.8 ± 0.3	11.2 ± 0.4	14.7	0.012
		Female (n = 9)	9.5 ± 0.3	10.4 ± 0.4	9.5	0.070
		Total (n = 17)	9.6 ± 0.2	10.8 ± 0.3	12.0	0.002
	TC (μg/mg of protein)	Male (n = 8)	37.5 ± 1.3	50.0 ± 5.0	33.5	0.030
		Female (n = 9)	40.5 ± 1.2	52.8 ± 6.7	30.5	0.107
		Total (n = 17)	39.1 ± 0.9	51.5 ± 4.2	31.8	0.009
	TG (μg/mg of protein)	Male (n = 8)	6.6 ± 1.2	5.6 ± 0.7	−15.6	0.469
		Female (n = 9)	4.8 ± 1.4	4.8 ± 0.7	−0.2	1.000
		Total (n = 17)	5.7 ± 0.9	5.2 ± 0.5	−8.7	0.643

Results are presented as the mean ± SEM (standard error of the mean). *p***^†^**, paired *t*-test performed for values obtained between Week 0 and 12. FI, fluorescence intensity (excitation = 370 nm, emission = 440 nm, 0.01 mg/dL of protein); HDL, high-density lipoprotein; TC, total cholesterol (μg/mg of protein); TG, triglyceride (μg/mg of protein); PON, paraoxonase; FRAP, ferric ion reduction ability.

**Table 6 pharmaceuticals-17-00132-t006:** Baseline characteristics of participants at Week 0 and physical activities during the 12 weeks ^1^.

Groups	At Week 0			During 12 Weeks				
	Age	BMI	MET	Total Exercise	Strength Exercise	Aerobic Exercise	MET ^2^	Physical Activity ^3^
	Mean ± SEM	Mean ± SEM	Score/Day	Min/wk	Min/wk	Min/wk	Score/Day	MET × 6 Day × 12 Weeks
Male (n = 8)	36.9 ± 1.9	31.0 ± 1.9	1.3 ± 0.1	655 ± 97	286 ± 9	390 ± 103	7.0 ± 0.3	504 ± 21
Female (n = 9)	38.4 ± 2.7	29.0 ± 1.4	1.1 ± 0.0	714 ± 95	194 ± 31	520 ± 84	7.4 ± 0.1	533 ± 11
Total (n = 17)	37.7 ± 1.6	30.1 ± 1.1	1.2 ± 0.1	687 ± 66	234 ± 21	459 ± 66	7.2 ± 0.1	518 ± 15

^1^ Data are expressed as the mean ± SEM (standard error of the mean). ^2^ The METs score was computed from the participant’s survey responses and categorized as follows: light (<3.0 METs), moderate (3.0~6.0 METs), and vigorous (>6.0 METs). ^3^ The total physical activity was calculated by MET score/day × 6 day × 12 weeks [42]. BMI, body mass index; MET, metabolic equivalents. wk, week.

## Data Availability

The data used to support the findings of this study are available from the corresponding author upon reasonable request.

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
