# Peer review of "Combination Therapy of Cuban Policosanol (Raydel®, 20 mg) and Intensive Exercise for 12 Weeks Resulted in Improvements in Obesity, Hypertension, and Dyslipidemia without a Decrease in Serum Coenzyme Q10: Enhancement of Lipoproteins Quality and Antioxidant Functionality in Obese Participants"

_pharmaceuticals, 2024, doi:10.3390/ph17010132_

Round 1
Reviewer 1 Report
Comments and Suggestions for Authors
The manuscript is interesting. The authors have presented the research findings nicely. The results of this study provide a basic understanding, indicating that in obese subjects, high-intensity exercise and Cuban policosanol intake may improve dyslipidemia and hypertension by enhancing HDL quality and antioxidant functionality without impairing CoQ10 metabolism. Here are a few suggestions for the authors: The authors should explain the reason for the selection of this dose of policosanol tablets (20 mg). The authors may improve the conclusion part.
Comments on the Quality of English Language
need improvement
Author Response
Thank you for your valuable comments and suggestions.
Please find attached doc as point-to-point response.

Reviewer 2 Report
Comments and Suggestions for Authors
The manuscript entitled „Combination therapy of Cuban policosanol (Raydel®, 20 mg) and intensive exercise for 12 weeks resulted in improvements in obesity, hypertension, and dyslipidemia without a decrease in serum coenzyme Q10: Enhancement of lipoproteins quality and antioxidant functionality in obese participants“ is well written and addresses an interesting topic.
Although the methodology is appropriate, there are significant limitations, as the authors themselves acknowledge, emphasizing the preliminary nature of the work. The lack of control groups, the small number of participants and the lack of a clear answer to the question of the independent effect of policosanol compared to exercise are serious shortcomings. Under other circumstances, I would probably recommend that this manuscript be rejected for publication.
The standard treatment for dyslipidemia includes statins and other medications in combination with moderate to intense physical activity and radical lifestyle modification. Intense exercise alone can have a large effect on increasing HDL-C levels, a moderate effect on lowering TG-rich lipoprotein levels, and a small effect on lowering total cholesterol and LDL-C levels. Statins are highly effective drugs and remain the cornerstone of modern guidelines for the treatment of dyslipidemia to reduce cardiovascular risk. However, statins are often associated with significant side effects. Therefore, the identification of a well-tolerated, side-effect-free substance that has some impact on treatment goals and cardiovascular disease prevention is highly desirable.
Policosanol, a natural product, has been extensively studied in this context as it is promoted under strong market pressure as a dietary supplement marketed as effective, natural and free of side effects. However, previous clinical trials have shown an unclear and insufficient effect of policosanol beyond placebo on lipoprotein levels in patients with hypercholesterolemia or hyperlipidemia. Further independent clinical studies are needed, as the authors themselves admit, and plan to conduct them.
Despite the preliminary nature of the results, I recommend the publication of this study in its current form with the following suggestion to the authors for further research: please include in future studies a phytochemical analysis of Cuban policosanol and compare it with similar substances from other sources, as the influence of an intrinsic factor (a constituent or ratio of a group of constituents agains a group of the others) peculiar to Cuban policosanol cannot be ruled out given the remarkable discrepancy between positive studies (mainly from Latin America) and less impressive results with policosanol from other sources and locations.
Author Response

(The authors gave the same response as above.)

Reviewer 3 Report
Comments and Suggestions for Authors
The reviewed manuscript related to experimental studies on the beneficial effects of Cuban policosanol is of scientific and practical significance and meets the scope of the journal.
The research is performed at a very high scientific level, involving various modern approaches, methods and techniques.
The results of the investigation are professionally interpreted, and statistical data are well processed, presented and evaluated.
It should be also mentioned here occurrence of the previous investigations, performed by the research team, in the given area and published in high-impact scientific journals.
However, it needs minor revision before publication.
1) Informed Consent Statement: Not applicable _ Informed consent is mandatory for all clinical trials involving human beings.
2) 2.3. Criteria of exclusion and inclusion : inclusion are not mentioned!
3) L. 251-253 should not be present in the Chapter "Materials and methods".
The intact charge and three-dimensional structure of LDL influence the electrophoretic mobility. Oxidized LDL with greater apo-B fragmentation exhibited faster migration toward the gel bottom, which is attributed to an increased negative charge.
4) Fig.5 : B and C do not correspond to the figure presentation, should be exchanged
B. Comparison of HDL3-associated PON-1 activity at weeks 0 and 12. C. Comparison of HDL2-associated FRA activity at weeks 0 and 12.
5) Figure 6. Mentioned on the graph (6C) levels of significance should be added into the figure caption.
6) "AO staining also showed that the CML alone group showed...."_Please use throughout the text for comfortable reading some synonyms to the verb Show, e.g. demonstrate, reveal etc ["show" occur 71 times within the text, demonstrate- 0 , reveal- 1] ; as well as for increase|decrease
7) In my personal opinion, it would be well to explain briefly (3-5 sentences) the basic items on the investigated drug: a summary of its composition (e.g. mixture of fatty alcohols....); countries where it is registered as a medicine;
main pharmacological effects|mechanisms caused by the active principles; already approved indications; why 12 weeks are taken as the administration course.
8) Some minor grammar and syntax mistakes occur, e.g.:
- measured all the anthropometric data and blood pressure data [presented!] in Table 2.
- the ministry of Health Care and Welfare (MOHW of Lorea.
- in according to the manufacturer’s protocol
- Detail procedures [!twice)
-earlier described methos
9) Abbreviations should be defined at first mention: e.g. PBS
Comments on the Quality of English LanguagePlease see the items 6 and 8 of my review.
Author Response

(The authors gave the same response as above.)
